# A New Mild Method for Synthesis of Marine Alkaloid Fascaplysin and Its Therapeutically Promising Derivatives

**DOI:** 10.3390/md21080424

**Published:** 2023-07-25

**Authors:** Oleg A. Tryapkin, Alexey V. Kantemirov, Sergey A. Dyshlovoy, Vladimir S. Prassolov, Pavel V. Spirin, Gunhild von Amsberg, Maria A. Sidorova, Maxim E. Zhidkov

**Affiliations:** 1Department of Chemistry and Materials, Institute of High Technologies and Advanced Materials, FEFU Campus, Far Eastern Federal University, Ajax Bay 10, Russky Island, 690922 Vladivostok, Russia; kantemirov_av@dvfu.ru (A.V.K.); sidorova_ma@dvfu.ru (M.A.S.); 2Department of Oncology, Hematology and Bone Marrow Transplantation with Section Pneumology, Hubertus Wald Tumorzentrum—University Cancer Center Hamburg (UCCH), University Medical Center Hamburg-Eppendorf, Martinistrasse 52, 20246 Hamburg, Germany; s.dyshlovoy@uke.de (S.A.D.); g.von-amsberg@uke.de (G.v.A.); 3Martini-Klinik Prostate Cancer Center, University Hospital Hamburg-Eppendorf, Martinistrasse 52, 20246 Hamburg, Germany; 4Department of Cancer Cell Biology, Engelhardt Institute of Molecular Biology, Russian Academy of Sciences, Vavilova 32, 119991 Moscow, Russia; prassolov45@mail.ru (V.S.P.); spirin.pvl@gmail.com (P.V.S.); 5Center for Precision Genome Editing and Genetic Technologies for Biomedicine, Engelhardt Institute of Molecular Biology, Russian Academy of Sciences, Vavilova 32, 119991 Moscow, Russia

**Keywords:** fascaplysin derivatives, synthesis, UV quaternization, DNA intercalation, prostate cancer, cytotoxicity

## Abstract

Fascaplysin is a marine alkaloid which is considered to be a lead drug candidate due to its diverse and potent biological activity. As an anticancer agent, fascaplysin holds a great potential due to the multiple targets affected by this alkaloid in cancer cells, including inhibition of cyclin-dependent kinase 4 (CDK4) and induction of intrinsic apoptosis. At the same time, the studies on structural optimization are hampered by its rather high toxicity, mainly caused by DNA intercalation. In addition, the number of methods for the syntheses of its derivatives is limited. In the current study, we report a new two-step method of synthesis of fascaplysin derivatives based on low temperature UV quaternization for the synthesis of thermolabile 9-benzyloxyfascaplysin and 6-*tert*-butylfascaplysin. 9-Benzyloxyfascaplysin was used as the starting compound to obtain 9-hydroxyfascaplysin. However, the latter was found to be chemically highly unstable. 6-*tert*-Butylfascaplysin revealed a significant decrease in DNA intercalation when compared to fascaplysin, while cytotoxicity was only slightly reduced. Therefore, the impact of DNA intercalation for the cytotoxic effects of fascaplysin and its derivatives needs to be questioned.

## 1. Introduction

Fascaplysin (**1**, Figure 1) is a pigment which was first isolated in 1988 from the marine sponge *Fascaplysinopsis* sp. It was the very first of a group of structurally related alkaloids based on the unique five-ring 12*H*-pyrido[1,2-*a*:3,4-*b′*]diindole system (**2**) [1]. Now, fascaplysin is considered a lead compound for the further development of novel drugs due to its broad spectrum of potent biological activities, including anticancer, antibacterial, antifungal, antiviral, and antimalarial activities [2,3,4,5,6]. Remarkably, anticancer activity has been shown in various cancer cell lines, including melanoma, breast, ovary, lung, leukemia, cervix, brain, and prostate cancer cells in vitro as well as in vivo in selected tumor models [7,8,9,10,11,12].

To date, several mechanisms of its action have been reported. One of the most studied and established is the selective inhibition of cyclin-dependent kinase 4 (CDK4), first reported by Sony et al. [13]. CDK4 regulates the G0-G1/S checkpoint of the cell cycle, and disruption of its function can lead to cancer [14,15]. Another important mechanism of fascaplysin cytotoxicity is the DNA intercalation, which is conditioned by a planar structure of the molecule [16]. In addition, a number of non-planar analogs of **1** were synthesized and studied. Among them, CDK4 inhibitors CA224 and BPT inhibited tubulin polymerization in vitro and showed antitumor activity in the colon cancer tumor model HCT-116 in vivo [17,18,19]. For other molecules, a high inhibitory activity was shown against CDK4, but in general, a significantly lower cytotoxicity against various tumor cell lines compared to the native alkaloid was shown in vitro [20,21,22,23,24,25]. Compound **1** also showed inhibitory activity against the oncogenic phosphatase inducer Cdc25B [26]. Lin et al. discovered that **1** inhibited angiogenesis via the suppression of vascular endothelial growth factor (VEGF) and the induction of apoptosis in the human umbilical vein endothelial cell (HUVEC) [27,28]. It also induced apoptosis in tumor cells by activating the tumor necrosis-related apoptosis-inducing ligand (TRAIL) signaling pathway, inducing a co-interaction between the apoptosis and autophagy pathways, which resulted in the death of the acute leukemia cells HL-60 and activated autophagy as a cytoprotective response through ROS and p8 in lymphoblastoid VEC cells [29,30,31]. Oh et al. found that fascaplysin inhibits TRKA and VEGFR2, as well as survivin and HIF-1α, resulting in inhibition of lung cancer cell growth. Fascaplysin has been shown to increase the phosphorylation of protein kinase B (PKB) and adenosine monophosphate-activated protein kinase (AMPK). Here, compound **1** was also found to synergize with a selective inhibitor of PI3K-AKT signaling, an AMPK inhibitor, and methotrexate [32,33]. Hamilton et al. investigated the cytotoxic effect of **1** against small cell and non-small cell lung cancer, wherein fascaplysin was found to induce tumor cell apoptosis through various mechanisms, including G1/0 cell cycle arrest and apoptosis. In addition, **1** was found to be synergistic with camptothecin, cisplatin, and afatinib [34,35,36]. Recently, Luo and Xu studied the anti-NSCLC effect of fascaplysin and, along with the known mechanisms, revealed the ability of fascaplysin to induce ferroptosis, as evidenced by increased levels of ROS and Fe^2+^ causing the downregulation of ferroptosis-associated protein and endoplasmic reticulum stress. Moreover, fascaplysin significantly upregulated the expression of PD-L1 in lung cancer cells, thereby improving the sensitivity of anti-PD-1 immunotherapy in vivo [37]. The discussion regarding other therapeutic targets of fascaplysin and anti-cholinesterase inhibitory activity should be noted [38]. Finally, Johnson et al. also found that **1** was a “balanced” opioid receptor agonist with a signaling profile reminiscent of that of endorphins [39].

At present, along with fascaplysin, a limited series of its derivatives has been studied (Figure 2). High anticancer activity was found for some halogen derivatives of **1**. Mono- and dibromo derivatives (**1a**–**c**) were effective against various tumor cell lines, including drug-resistant prostate cancer and glioblastoma [40,41,42,43]. In addition, 3,10-dibromofascaplysin (**1c**) induced apoptosis in leukemic cells and had a synergistic effect with cytarabine [44]. Chlorofascaplysin **1d** exhibited anticancer and anti-angiogenesis effects in breast cancer cells, inhibited the tumor growth in an Ehrlich solid tumor model in mice and showed no apparent toxicities in experimental tumor mice at therapeutic doses [45]. For some halogen derivatives, the ability to affect P-gp was also studied; this study showed that the activity and effect on cell survival for the difluoro-derivative **1e** was higher [46]. In another study, the methyl derivative of **1** at C-9 (**1f**) was found to have a greater multitarget effect against Alzheimer’s disease than the native alkaloid, including better inhibition of Aβ aggregation [47,48]. Derivatives of compound **1** at C-9 (**1g**–**i**) were also demonstrated to have superior antibacterial activity against methicillin-resistant *Staphylococcus aureus* (MRSA) and *Escherichia coli* in vitro [49,50]. Thus, for some fascaplysin derivatives a therapeutic potential has been shown; however, the number of studied derivatives is rather small due to the limited number of synthetic methods available at present.

Over the past 30 years, more than 10 different approaches to the syntheses of fascaplysin and its derivatives have been developed. The first total synthesis of compound **1** was carried out by Pelcman and Gribble from indole in seven steps. The approach involved the formation of diindole, which was then cyclized, dehydrogenated, and oxidized to **1** [51]. Rocca et al. developed a shorter four-step route for the synthesis of **1** starting from *N*-substituted 2-aminophenylboronic acid and 4-iodo-3-fluoropyridine [52]. Zhidkov et al. elaborated the three-step strategy to compound **1**, including the Fischer cyclization between 10-methyl-7,8-dihydro-6H-pyrido[1,2-a]indol-9-one and phenylhydrazine followed by oxidation and the Baeyer–Villiger rearrangement of the intermediate [53]. Waldman et al. proposed a two-stage method for the synthesis of **1** from an *N*-protected acetylene derivative of formylindole with a silver catalyst in a microwave reactor [54]. Another approach involved the condensation of indigo with methylene active esters and the subsequent reduction and hydrolysis of the intermediates to fascaplysin and several of its derivatives at the central ring [55,56]. However, the most productive strategy was based on the syntheses of 1-benzoyl-β-carbolines as the key intermediates. Molina et al. first proposed a four-stage method for the preparation of **1** starting from an iminophosphorane derivative of indole [57]. Then, this approach was further improved by Radchenko, who suggested using the high-temperature cyclization of o-halo-substituted 1-benzoyl-β-carbolines to obtain the skeleton of fascaplysin, as well as a method for the obtaining of such intermediates in four steps [58]. Zhidkov proposed a synthesis of 1-(2-fluorobenzoyl)-β-carboline by the Minisci homolytic acylation of unsubstituted β-carboline with *o*-fluorobenzaldehyde under microwave irradiation [59]. Bharate et al. elaborated the method for the obtaining of substituted benzoyl-β-carboline by tandem condensation between substituted phenylglyoxal and tryptamine [38]. Zhu et al. managed to combine that sequence of transformations to a one-pot cascade coupling protocol that included the sequential iodination of the corresponding acetophenone, the Kornblum oxidation of the intermediate in the presence of DMSO to phenylglyoxal, and its Pictet–Spengler condensation with tryptamine, followed by the oxidation of the intermediate [60]. Battini et al. proposed to carry out this reaction without the use of H_2_O_2_ as a co-oxidizer with comparable yields [61]. Later, Dighe et al. suggested using phenylacetylenes as an alternative to substituted acetophenones for the synthesis of **1** [62]. Thus, currently, the most convenient and effective two-stage method for the obtaining of fascaplysin and its derivatives is the production of substituted 1-benzoyl-β-carbolines according to Zhu and their high-temperature cyclization using Radchenko’s protocol. Although this approach provides some variety of derivatives of **1**, its significant limitation is the possibility of obtaining exclusively thermostable derivatives of the alkaloid.

In this work, we developed a new method for the quaternization of substituted 1-benzoyl-β-carbolines under the action of UV irradiation at low temperature and demonstrated its capabilities by synthesizing 9-benzyloxyfascaplysin (**3**) and 6-*tert*-butylfascaplysin (**4**), which could not be otherwise be obtained by the high-temperature quaternization. For compound **4**, the ability to intercalate into DNA and its correlation with cytotoxicity against prostate cancer were evaluated. In addition, the conditions for the conversion of compound 3 to 6-hydroxyfascaplysin, a promising lead compound for the further synthesis of conjugates for targeted drug delivery, were also studied.

## 2. Results and Discussion

### 2.1. Development of the UV Quaternization Protocol

The starting point for our research was based on the known ability of papaveraldine (**5a**) and papaverine-like 1-(3-isopropoxybenzoylbenzoyl)-6,7-dimethoxyisoquinoline (**5b**) to convert to the corresponding products **6a**–**b** with the storing of their solutions in the light (Figure 1) [63,64,65].

In order to study the possibility of the similar reaction with 1-benzoyl-β-carbolines, the variety of starting materials was prepared according to the combined methods of Zhu et al. and Battini et al. from tryptamine (**7**) and acetophenones **8**–**13** (compounds **14**–**19**) (Figure 2) [60,61]. The use of H_2_O_2_ as a co-oxidizer, as in the original method, led to the formation of by-products and reduced the yields of target 1-benzoyl-β-carbolines. The usage of an equimolar amount or an excess of I_2_, as in the Battini methodology, resulted in the formation of the product of the iodination of 1-benzoyl-β-carboline at C-6 and also reduced the yield of the target product.

Initially, based on the analogy with isoquinolines, we tried to carry out the target transformation under comparable conditions for unsubstituted 1-benzoyl-β-carboline (**14**), but the product of quaternization was not observed. Increasing and decreasing acidity to facilitate quaternization was not successful. The usage of derivatives with electron donor (**15**) and electron-withdrawing (**16**) substituents at the *meta*-position of the benzoyl fragment demonstrated the formation of the corresponding products in traces. To increase the yield of the product, we tried to use various solvents, but without success. The only exception was acetonitrile, but in this case, the product was also formed in trace amounts. An increase in the reaction temperature in the DMSO led to the formation of side products. The addition of radical reaction catalysts such as 2,2′-azobis(2-methylpropionitrile) (AIBN) or benzoyl peroxide (BPO) also resulted in the formation of by-products. Based on this, it seemed appropriate to lower the temperature of the reaction and also to replace the hydrogen atom in the *ortho*-position of the benzoyl fragment of β-carboline with halogen atoms. In this case, an increase in the yield of the reaction product was expected in the following order: chlorine, bromine, iodine. The use of the chlorine derivative (**18**) did not lead to the formation of the product. The usage of the derivative with bromine (**17**) proved to be successful and made it possible to obtain the target product in a 10% yield. The application of the iodine derivative (**19**) increased the yield of compound **1** to 50%. The conditions used are summarized in Table 1.

In case **19**, the usage of acetonitrile with decreasing temperature to hinder side reactions made it possible to obtain **1** in the same yield. In the reaction mixture, only initial compound **19** and quaternization product **1** were observed. That made it possible to implement such a protocol for carrying out the reaction, during which the isolation of the product and the repeated irradiation of the reaction mixture alternated successively. As a result, the yield of the target product was increased to 91% (Figure 3).

### 2.2. Synthesis of 9-Benzyloxyfascaplysin

The developed approach was applied to the synthesis of 9-benzyloxyfascaplysin (**3**). 5-Benzyloxy-3-formylindole (**21**) was obtained by the Vilsmeier–Haack acylation of 5-benzyloxyindole (**20**) with phosphorus oxychloride in DMF. Then, the interaction of **21** with nitromethane allowed the preparation of 5-benzyloxy-3-(2-nitroethenyl)indole (**22**) in an 83% yield. The reduction of **22** led to the formation of 5-benzyloxytryptamine (**23**). The substituted 1-benzoyl-6-benzyloxy-β-carbolines **24** and **25** were synthesized from tryptamine **23** and the 2-substituted acetophenones **11** and **13** by the combined method of Zhu et al. and Battini et al. [60,61] (Figure 4).

The UV quaternization of 6-benzyloxy-1-(2′-iodobenzoyl)-β-carboline (**25**) resulted in product **3** with a yield of 88% (Figure 5). The high-temperature quaternization of 6-benzyloxy-1-(2′-bromobenzoyl)-β-carboline (**24**) led to the formation of a mixture of products, among which **3** was not found (Table 2).

The removal of the benzyl group, which was expected to yield 9-hydroxyfascaplysin (**26**), instead resulted in the formation of what was presumably quinoid compound **27**. It was proposed based on the HRMS data, and its obtainment can be explained by the tendency towards oxidation of the fragment of *p*-hydroxyaminophenol of compound **26** (Figure 6). The resulting product turned out to be extremely poorly soluble; thus, it did not allow us to rigorously prove its structure or to study its biological properties and possibilities for derivatization. Apparently, these observations suggest that there is a need to obtain the hydroxy derivatives of fascaplysin at other positions.

### 2.3. Study of 6-tert-Butylfascaplysin

#### 2.3.1. Chemistry

For the syntheses of 2′-substituted 1-benzoyl-3-*tert*-butyl-β-carbolines **36** and **37**, it was necessary to use the Franklin and White strategy for the syntheses of α-substituted tryptamines [66]. At the first stage, compound **30** was obtained by aldol condensation between isatin (**28**) and 3,3-dimethyl-2-butanone (pinacoline, **29**). Due to retro-aldol decay as a result of the interaction of aldol **30** with hydroxylamine, two additional steps of dehydration and the subsequent reduction of the resulting double bond were added. The direct reduction of the obtained oxime **33** to α-*tert*-butyltryptamine (**35**) under the action of various reagents did not lead to the desired results. Tryptamine **35** was obtained by the successive reduction of the first oxime fragment under the action of hydrogen over PtO_2_ and then by the reduction of the lactam fragment by the system BH_3_·THF. The resulting crude product was introduced to the one-pot cascade coupling protocol developed by Zhu et al. [60], which finally made it possible to obtain the target compounds **36** and **37** (Figure 7).

The low-temperature quaternization under UV irradiation of 1-(2′-iodo-benzoyl)-6-*tert*-butyl-β-carboline (**37**) made it possible to obtain the target 6-*tert*-butylfascaplysin (**4**) in a 92% yield (Figure 8). The UV quaternization of its analog **36** in DMSO for 9 h resulted in a 30% yield of **4**, which was better than that for the preparation of fascaplysin from compound **17** (Table 1). However, increasing the irradiation time still led to the formation of by-products. Numerous attempts to prepare derivative **4** by the high-temperature quaternization of β-carboline **36** failed (Table 3).

#### 2.3.2. Biological Studies

As the intercalation into DNA is considered an important component of the biological mechanism of action of fascaplysin, we evaluated and compared this activity for the synthesized 6-*tert*-butylfascaplysin (**4**) and unsubstituted fascaplysin (**1**). The drug–DNA complex formation was assessed by a fluorescent intercalator displacement assay based on the detachment of the thiazole orange (TO) intercalating agent from the DNA duplex by the tested compounds. The half-maximal concentrations for the effective displacement of TO from the DNA complex were calculated and are presented in Figure 3. In this experiment, the well-established DNA-binding compound, propidium iodide (PI), was used as a positive control. We found that compound **4** was 7-fold less effective at displacing TO from the fluorescent complex with DNA than **1** (Figure 3).

To investigate the effect of the *tert*-butyl group on anticancer activity and selectivity, we evaluated the cytotoxic activity of unsubstituted fascaplysin (**1**) and 6-*tert*-butylfascaplysin (**4**) in various cancer and non-cancer cell lines using the well-established MTT method (Table 4). The effects on the human prostate cancer cells PC3 and DU145 (hormone-independent cells), 22Rv1 (partially hormone-independent cells), and LNCaP (hormone-sensitive cells), as well as on the normal (non-cancerous) human cells PNT2, MRC-9, and HEK293, were investigated. The mean cytotoxicity of compound 4 towards either cancerous or non-cancerous cells decreased by ~2-fold compared to that of fascaplysin (**1**). At the same time, the selectivity indexes of both compounds were comparable. In general, the absolute cytotoxicity of compound **4** remained rather high since IC_50_ did not exceed 2 μM in either cell line.

Of note, 6-*tert*-butylfascaplysin (**4**) revealed a decreased ability to intercalate into DNA due to steric obstacles (i.e., introduction of 6-*tert*-butyl group). However, this did not result in a meaningful decrease in its cytotoxicity. These observations suggest that an intercalation into DNA may not be the main mechanism mediating the cytotoxicity of fascaplysin derivatives, as has previously been postulated.

## 3. Materials and Methods

### 3.1. Chemistry

All of the starting materials are commercially available. Commercial reagents were used without any purification. For UV irradiation, the high-pressure mercury UV lamp DRT-1000 was used. The products were isolated by MPLC: Buchi B-688 pump; glass column C-690 (15 × 460 mm) with silica gel (particle size 0.015–0.040 mm); and UV detector Knauer K-2001. The analytical examples were purified by the Shimadzu HPLC system (model: LC-20AP) equipped with a UV detector (model: SPD 20A), using a Supelco C18 (5 µm, 20 × 250 mm) column using the MeOH:H_2_O (20:80, 50:50, 70:30) mobile phase by isocratic elution at a flow rate of 15 mL/min. The NMR spectra were recorded with an NMR instrument operating at 400 MHz (^1^H) and 100 MHz (^13^C). Proton spectra were referenced to TMS as an internal standard and, in some cases, to the residual signal of used solvents. Carbon chemical shifts were determined relative to the ^13^C signal of TMS or the used solvents. Chemical shifts are given on the δ scale (ppm). Coupling constants (J) are given in Hz. Multiplicities are indicated as follows: s (singlet), d (doublet), t (triplet), q (quartet), m (multiplet), or br (broadened). The original spectra of the relative compounds can be found in Appendix A. High-resolution mass spectra (HRMS) were obtained with a time-of-flight (TOF) mass spectrometer (model Agilent TOF 6210) equipped with an electrospray source at atmospheric pressure ionization (ESI).

#### 3.1.1. Synthesis of Compound **21**

POCl_3_ (0.59 mL, 6.3 mmol) was added dropwise with stirring to 2 mL of DMF cooled to 0 °C. Then, a solution of 1.00 g (4.5 mmol) of **20** in 1.5 mL of DMF was added dropwise, avoiding heating above 15 °C. The mixture was stirred while heating to 35 °C for 1.5 h. Afterwards, the mixture was poured into a 10 mL mixture of H_2_O with ace. Then, NaOH (aq) was added to an alkaline pH, and the mixture was refluxed for 1 h. The precipitate was filtered off and washed until neutral and dried. The resulting product was a beige powder (1.01 g, 90%, melting point 235–237 °C, 234–235 °C according to the literature [67]).

#### 3.1.2. Synthesis of Compound **22**

A mixture of 0.99 g (4.0 mmol) of **21**, 0.32 g (4.0 mmol) of CH_3_COONH_4_, and 6 mL of CH_3_NO_2_ was refluxed for 1 h. After cooling to room temperature, 4 mL of acetone and 100 mL of H_2_O were added. The precipitate that formed was filtered off and dried. The resulting product was orange crystals (0.96 g, 83%, melting point 175–177 °C, 179–180 °C according to the literature [67]).

#### 3.1.3. Synthesis of Compound **23**

A mixture of NaBH_4_ (0.72 g, 19.3 mmol), 53 mL of THF, and BF_3_·OEt_2_ (3 mL, 24.3 mmol) were placed in a dry flat-bottomed 100 mL flask with a septum and a rotor cooled to 0 °C. The solution was then stirred at room temperature for 15 min. A solution of **22** (0.96 g, 3.6 mmol) in 11 mL of THF was added dropwise through a septum with a syringe. Then, the mixture was refluxed for two hours. After cooling to room temperature, the mixture was poured into a 500 mL flask; ice was added until the reaction stopped, and 300 mL of H_2_O was added. A solution of 1M HCl was added to the mixture to an acidic pH; then, it was refluxed for two hours. After cooling to room temperature, the mixture was washed with ether, NaOH (aq) was added to a slightly basic pH, and the mixture was extracted with ether. The extract was dried, evaporated under reduced pressure, and immediately introduced to the next reaction. Product **23** was not isolated or characterized due to low stability.

#### 3.1.4. Preparation of Compound **30**

Pinacoline (**29**) (6.00 g, 59.9 mmol) and NaOH (1.15 g, 28.8 mmol) in 5 mL of H_2_O were successively added to a suspension of isatin (**28**) (5.00 g, 34.0 mmol) in EtOH (100 mL). Next, the reaction mixture was stirred on a magnetic stirrer heated to 50 °C. The reaction mixture was kept at a constant weak alkaline pH for 24 h. The progress of the reaction was monitored by TLC. After the appearance of traces of the by-product, the reaction was complete. Then, the mixture was evaporated under reduced pressure, and the residue was recrystallized from H_2_O. The crystals were filtered off, washed with cold H_2_O and dried. The obtained product was light beige crystals (2.30 g, 30%).

^1^H NMR (400 MHz, CDCl_3_): *δ* 8.46 (s, 1H), 7.30 (d, *J* = 7.4 Hz, 1H), 7.24 (td, *J* = 7.8, 1.0 Hz, 1H), 7.02 (td, *J* = 7.5, 0.6 Hz, 1H), 6.88 (d, *J* = 7.7 Hz, 1H), 4.82 (s, 1H), 3.33 (d, *J* = 17.6 Hz, 1H), 3.05 (d, *J* = 17.6 Hz, 1 H), 1.08 (s, 9H). ^13^C NMR (100 MHz, CDCl_3_): δ 215.3, 178.6, 140.7, 130.4, 129.9, 124.0, 123.0, 110.5, 74.8, 44.6, 42.6, 25.9. HRMS-ESI, *m/z*: [M + H]^+^ calculated for C_14_H_18_NO_3_^+^ 248.1281, obtained 248.1291.

#### 3.1.5. Preparation of Compound **31**

Compound **30** (2.00 g, 8.1 mmol) was dissolved in 5 mL of CH_3_COOH, and 4 drops of HCl (aq) were added to the solution. The reaction mixture was stirred on a magnetic stirrer heated to 75 °C. The progress of the reaction was monitored by TLC. Then, the mixture was poured into H_2_O and neutralized with NaHCO_3_; the solution was extracted with EtOAc (3 × 20 mL); and the extract was dried and evaporated under reduced pressure. The obtained product was orange crystals (1.84 g, 99%).

^1^H NMR (400 MHz, CDCl_3_): *δ* 8.36 (d, 7.8 Hz, 1H), 8.21 (br. s, 1H), 7.46 (s, 1H), 7.32 (td, *J* = 7.7, 0.9 Hz, 1H), 7.02 (td, *J* = 7.7, 0.8 Hz, 1H), 6.86 (d, *J* = 7.7 Hz, 1H), 1.29 (s, 9H). ^13^C NMR (100 MHz, CDCl_3_): *δ* 206.5, 169.5, 143.1, 136.1, 132.6, 128.1, 125.7, 122.9, 120.7, 110.0, 44.8, 26.2. HRMS-ESI, *m/z*: [M + H]^+^ calculated for C_14_H_16_NO_2_^+^ 230.1176, obtained 230.1193.

#### 3.1.6. Preparation of Compound **32**

Compound **31** (0.98 g, 4.3 mmol) was dissolved in 30 mL of MeOH. A catalytic amount of 10% Pd/C was added to the solution. The reaction was carried out in a hydrogen atmosphere (6 bar). The mixture was stirred at room temperature for 18 h. The bright orange color of the solution disappeared. The mixture was then poured into H_2_O and extracted with EtOAc (3 × 20 mL). The extract was dried and evaporated under reduced pressure. The obtained product was light yellow crystals (0.93 g, 94%).

^1^H NMR (400 MHz, CDCl_3_): *δ* 8.30 (br. s, 1H), 7.19 (t, *J* = 7.7 Hz, 1H), 7.09 (d, *J* = 7.4 Hz, 1H), 6.97 (td, *J* = 7.3, 0.6 Hz, 1H), 6.88 (d, *J* = 7.8 Hz, 1H), 3.91 (dd, *J* = 8.5, 3.2 Hz, 1H), 3.31 (dd, *J* = 18.4, 3.3 Hz, 1H), 3.00 (dd, *J* = 18.3, 8.6 Hz 1H), 1.17 (s, 9H). ^13^C NMR (100 MHz, CDCl_3_): *δ* 212.8, 180.0, 141.4, 129.7, 128.0, 124.2, 122.4, 109.6, 43.9, 41.4, 38.1, 26.4. HRMS-ESI, *m/z*: [M + H]^+^ calculated for C_14_H_18_NO_2_^+^ 232.1332, obtained 232.1332.

#### 3.1.7. Preparation of Compound **33**

Finely ground NH_2_OH·HCl (1.50 g, 21.5 mmol) and CH_3_COONa (1.76 g, 22.0 mmol) were added to a flat-bottomed flask. Next, 30 mL of MeOH was added, followed by the introduction of compound **32** (1.28 g, 5.2 mmol) into the mixture. The reaction mixture was stirred on a magnetic stirrer heated to 40 °C for 24 h. Then, the same amount of NH_2_OH·HCl and CH_3_COONa was added to the mixture, which was stirred with heating for another 24 h. At the end, the mixture was diluted with H_2_O and extracted with EtOAc (3 × 30 mL). The extract was dried and evaporated. The resulting product was a light cream powder (1.25 g, 97%).

^1^H NMR (400 MHz, CDCl_3_): *δ* 8.84 (br. s, 1H), 8.46 (s, 1H), 7.24 (d, *J* = 7.5 Hz, 1H), 7.19 (t, *J* = 7.8 Hz, 1H), 6.99 (td, *J* = 7.5, 0.5 Hz, 1H), 6.86 (d, *J* = 7.6 Hz, 1H), 4.43 (t, *J* = 8.6 Hz, 1H), 2.99 (dd, *J* = 13.9, 7.8 Hz, 1H) 2.68 (dd, *J* = 14.0, 9.6 Hz, 1H), 1.09 (s, 9H). ^13^C NMR (100 MHz, CDCl_3_): *δ* 180.2, 163.9, 141.2, 129.5, 128.0, 125.3, 122.2, 109.4, 42.1, 37.7, 28.0, 27.2. HRMS-ESI, *m/z*: [M + H]^+^ calculated for C_14_H_19_N_2_O_2_^+^ 247.1441, obtained 247.1454.

#### 3.1.8. Synthesis of Compound **35**

Compound **33** (0.74 g, 3.0 mmol) was dissolved in 20 mL of MeOH. A catalytic amount of PtO_2_ was added to the solution. The reaction was carried out in a hydrogen atmosphere (6 bar). Thereafter, the mixture was stirred at room temperature for 48 h. Afterwards, the precipitate that formed was filtered off and dried. Due to its instability, the resulting product **34** was immediately introduced to the next stage of the synthesis. NaBH_4_ (0.57 g, 15.1 mmol) was added to 10 mL of freshly distilled THF, which was in a flat-bottomed conical flask with a stirrer. The flask was placed in an ice bath; its contents were cooled to 0 °C, after which freshly distilled BF_3_·OEt_2_ (2.043 mL, 16.6 mmol) was added in portions. The ice bath was removed, and the reaction mixture was stirred for another 15 min at room temperature. The previously obtained product **34** was added to the solution. The mixture was heated to reflux. After 2 h, the mixture was cooled to room temperature, after which a 10% HCl solution was added to it to a fivefold dilution. The mixture was again heated to reflux and after 2 h was cooled to room temperature. The mixture was then neutralized with Na_2_CO_3_ (aq) and extracted with EtOAc (3 × 30 mL). The extract was dried, evaporated under reduced pressure, and immediately introduced to the next reaction. Product **35** was not isolated or characterized due to low stability.

#### 3.1.9. Preparation of Substituted 1-Benzoyl-β-Carbolines **14**–**19**, **24**–**25**, **36**–**37**

Corresponding acetophenone (0.5 mmol) and iodine (0.09 g, 0.4 mmol) were added to 2 mL of DMSO, and the resulting solution was heated at 110 °C for 1 h. Afterwards, tryptamine or its derivative (0.5 mmol) was added to the solution, and this solution was stirred at the same temperature for 3–4 h until the completion of the reaction (monitored by TLC). Then, the reaction mixture was cooled to room temperature followed by the addition of H_2_O (50 mL) and extraction with EtOAc (2 × 25 mL). The extract was washed with 10% Na_2_S_2_O_3_, dried over Na_2_SO_4_, filtered, and evaporated under reduced pressure. The residue was purified by MPLC using benzene or hexane/benzene as an eluent to give the desired product.

For compound **14**: yellow solid, 35%. The spectral data correspond to the literature [62].

For compound **15**: yellow solid, 38%. The spectral data correspond to the literature [62].

For compound **16**: yellow solid, 31%. The spectral data correspond to the literature [62].

For compound **17**: yellow solid, 42%. The spectral data correspond to the literature [46].

For compound **18**: yellow solid, 38%. The spectral data correspond to the literature [62].

For compound **19**: yellow solid, 37%. The data correspond to the literature [60].

For compound **24**: orange solid, 28%. ^1^H NMR (400 MHz, CDCl_3_): *δ* 10.35 (br. s, 1H), 8.55 (d, *J* = 5.0 Hz, 1H), 8.13 (d, *J* = 5.0 Hz, 1H), 7.71 (m, 2H), 7.59–7.36 (m, 10H), 5.23 (s, 2H). ^13^C NMR (100 MHz, CDCl_3_): *δ* 197.8. 153.6, 140.1, 138.1, 137.1, 136.7, 135.9, 135.0, 132.8, 131.4, 130.9, 129.5, 128.4, 127.8, 127.3, 126.6, 120.9, 119.8, 119.6, 118.9, 112.6, 105.3, 70.7. HRMS-ESI, *m/z*: [M + H]^+^ calculated for C_25_H_18_^79^BrN_2_O_2_^+^ 457.0546, obtained 457.0547.

For compound **25**: orange solid, 31%. ^1^H NMR (400 MHz, CDCl_3_): *δ* 10.32 (br. s, 1H), 8.49 (d, *J* = 5.0 Hz, 1H), 8.08 (d, *J* = 5.0 Hz, 1H), 7.94 (d, *J* = 8.0 Hz, 1H), 7.68 (d, *J* = 2.2 Hz, 1H), 7.53–7.45 (m, 5H), 7.40 (t, *J* = 7.4 Hz, 2H), 7.35–7.32 (m, 2H), 7.19 (td, *J* = 7.5, 2.1 Hz, 1H), 5.18 (s, 2H); ^13^C NMR (100 MHz, CDCl_3_): 198.9, 153.7, 143.7, 139.3, 138.0, 137.4, 136.7, 135.9, 134.5, 131.4, 130.9, 129.2, 128.4, 127.8, 127.3, 127.2, 120.9, 119.6, 118.8, 115.7, 112.6, 105.3, 92.6, 70.8. HRMS-ESI, *m/z*: [M + H]^+^ calculated for C_25_H_18_IN_2_O_2_^+^ 505.0408, obtained 505.0426.

For compound **36**: yellow solid, 5%. ^1^H NMR (400 MHz, CDCl_3_): *δ* 10.22 (br. s, 1H), 8.21 (s, 1H), 8.18 (d, *J* = 7.8 Hz, 1H), 7.95 (d, *J* = 7.9, 1H), 7.60–7.56 (m, 3H), 7.48 (t, *J* = 7.5 Hz, 1H), 7.36–7.32 (m, 1H), 7.20 (td, *J* = 7.7, 1.5 Hz, 1H), 1.35 (s, 9H). ^13^C NMR (100 MHz, CDCl_3_): *δ* 199.3, 158.3, 144.3, 141.4, 139.1, 135.4, 132.9, 132.5, 130.8, 129.9, 129.0, 127.0, 121.7, 121.1, 120.6, 114.5, 112.0, 93.2, 37.6, 30.6. HRMS-ESI, *m/z*: [M + H]^+^ calculated for C_22_H_20_^79^BrN_2_O^+^ 407.0754, obtained 407.0768.

For compound **37**: yellow solid, 6%. ^1^H NMR (400 MHz, CDCl_3_): *δ* 10.20 (br. s, 1H), 8.20 (s, 1H), 8.18 (dd, *J* = 8.0, 0.5 Hz, 1H), 7.67 (d, *J* = 8.0, 0.8 Hz, 1H), 7.61–7.58 (m, 3H), 7.44 (td, *J* = 7.5, 1.0 Hz, 1H), 7.38–7.31 (m, 2H), 1.35 (s, 9H). ^13^C (100 MHz, CDCl_3_): *δ* 198.2, 158.4, 141.4, 140.8, 135.1, 133.4, 132.6, 132.5, 130.7, 130.2, 129.0, 126.3, 121.7, 121.1, 120.6, 120.4, 114.4, 111.9, 37.5, 30.5. HRMS-ESI, *m/z*: [M + H]^+^ calculated for C_22_H_20_IN_2_O^+^ 455.0615, obtained 455.0624.

#### 3.1.10. Preparation of Fascaplysins **1**, **3**, **4**

A solution of the corresponding 1-benzoyl-β-carboline (0.05 mmol) in 10 mL of acetonitrile was irradiated with UV for 30–90 min. The solution was evaporated, the residue was washed from the starting β-carboline with acetonitrile or benzene, depending on the solubility of the resulting fascaplysin. For re-irradiation, the non-reacted β-carboline solution was evaporated, dissolved in 10 mL of acetonitrile, and irradiated again. The end of the reaction was monitored by TLC. After filtration, the fascaplysin was washed with EtOH, then evaporated under reduced pressure and dried. Then, the product was dissolved in H_2_O, and an aqueous solution of Na_2_CO_3_ was added. The resulting dark green precipitate of the deprotonated form of the product was filtered, washed with water, and washed off with an aqueous solution of HCl. The resulting solution was evaporated and dried.

For fascaplysin (**1**): red solid, 91%. The data correspond to the literature [4].

For 9-benzyloxyfascaplysin (**3**): red solid, 88%. ^1^H NMR (400 MHz, CD_3_OD): *δ* 9.26 (d, *J* = 6.2 Hz, 1 H), 8.87 (d, *J* = 6.2 Hz, 1 H), 8.27 (d, *J* = 8.1 Hz, 1 H), 7.97–8.03 (m, 2 H), 7.92 (t, *J* = 7.6 Hz, 1 H), 7.73–7.67 (m, 2 H), 7.57 (dd, *J* = 9.0, 2.2 Hz, 1 H), 7.48 (d, *J* = 7.3 Hz, 2H), 7.33–7.40 (m, 2 H), 7.30 (d, *J* = 7.2 Hz, 1 H), 5.22 (s, 2 H). ^13^C NMR (100 MHz, CD_3_OD): *δ* 181.6, 155.2, 147.1, 142.6, 142.3, 140.5, 136.5, 136.4, 130.8, 127.9, 127.4, 127.1, 125.8, 125.0, 124.8, 123.8, 122.1, 120.1, 119.3, 115. 7, 114.6, 113.9, 106.7, 104.9, 70.1. HRMS-ESI, *m/z*: [M]^+^ calculated for C_25_H_17_N_2_O_2_^+^ 377.1285, obtained 377.1284.

For 6-*tert*-butylfascaplysin (**4**): red solid, 92%. ^1^H NMR (400 MHz, CD_3_OD): *δ* 9.00 (s, 1H), 8.53 (d, *J* = 8.0 Hz, 1H), 8.43 (d, *J* = 8.6 Hz, 1H), 8.08 (d, *J* = 6.8 Hz, 1H), 7.96 (t, *J* = 7.6 Hz, 1H), 7.85 (t, *J* = 7.6 Hz, 1H), 7.77–7.71 (m, 2H), 7.48 (t, *J* = 7.5 Hz, 1H), 1.95 (s, 9H). ^13^C NMR (100 MHz, MeOH-d4): *δ* 182.9, 149.1, 148.8, 141.5, 136.1, 134.7, 131.5, 130.5, 125.5. 125.3, 124.8, 124.4, 124.2, 122.9, 119.6, 119.2, 113.2, 112.0, 36.6, 30.4. HRMS-ESI, *m/z*: [M]^+^ calculated for C_22_H_19_N_2_O^+^ 327.1492, obtained 327.1486.

#### 3.1.11. An Attempt to Remove the Benzyl Protection from **3**

Method A. Freshly distilled BF_3_·OEt_2_ (0.02 mL, 1.5 mmol) in 2 mL of dry acetonitrile was slowly added to a stirred solution of **3** (5 mg, 0.01 mmol) and anhydrous NaI (2 mg, 0.1 mmol) in 5 mL of dry acetonitrile at 0 °C for 10–15 min. The mixture was stirred at 0 °C for 2 h and then for another 8 h at room temperature. The reaction was monitored by TLC. The formation of a rapidly oxidizing compound, presumably compound **27**, with by-products was observed.

Method B. To 5 mg of 9-benzyloxyfascaplysin (0.01 mmol) in 5 mL of ethanol in a 10 mL flat-bottom flask, a catalytic amount of 10% palladium on carbon was added; this was sealed with a septum and left to stir under a hydrogen atmosphere on a magnetic stirrer at room temperature for 16 h. The reaction mixture changed color to green after 15 min. The progress of the reaction was monitored by TLC. The formation of a rapidly oxidizing compound, presumably compound **27**, with by-products was observed.

HRMS-ESI, *m/z*: M^+^ calculated for C_18_H_9_IN_2_O_2_^+^ 285.0659, obtained 285.0672.

### 3.2. Biological Assay

#### 3.2.1. Reagents

Propidium iodide and MTT reagent (3-(4,5-dimethylthiazol-2-yl)-2,5-diphenyltetrazolium bromide) were purchased from Sigma (Taufkirchen, Germany); thiazole orange was purchased from Merck (Darmstadt, Germany); RNase was purchased from Carl Roth (Karlsruhe, Germany).

#### 3.2.2. Cell Lines and Culture Conditions

Cell lines PC-3, DU145, 22Rv1, and LNCaP (human prostate cancer), as well as PNT2 (human prostate non-cancer), were purchased from ATCC (Manassas, VA, USA). Human embryonic kidney cells HEK 293T and human fibroblasts MRC-9 cell lines were purchased from ECACC (Salisbury, UK). The cells used had a passage ≤ 30 and were recently authenticated by Multiplexion GmbH (Heidelberg, Germany). The cells were cultured at 37 °C as monolayers in a humidified atmosphere of 5% CO_2_. The culture mediums used for cultivation and experiments: for PNT2, LNCaP, 22Rv1, PC-3, and DU145 cells—RPMI Glutamax^TM^-I medium (gibco^®^ Life Technologies^TM^, Paisley, UK), supplemented with 10% FBS (gibco^®^ Life Technologies^TM^) and 1% penicillin/streptomycin (Invitrogen, Waltham, MA, USA); for MRC-9 and HEK 293 cells—DMEM GlutamaxTM-I medium (gibco^®^ Life Technologies^TM^), supplemented with 10% FBS and 1% penicillin/streptomycin.

#### 3.2.3. MTT Assay

The cytotoxic activity of the synthesized compounds was examined using an MTT assay, which was performed as previously reported [68]. In brief, the cells were seeded in 96-well plates (6000 cells/well) and incubated overnight. The medium was replaced with fresh medium containing the tested drugs, and the cells were incubated for an additional 48 h. Next, 10 μL/well of 5 mg/mL MTT reagent was added, and the cells were incubated for an additional 2 h. Then, the medium was removed, the formazan crystals were dried overnight, and DMSO was added to each well (50 μL/well). The absorbance of the DMSO solutions was measured using a TECAN Infinite F200PRO reader (Männedorf, Switzerland). IC_50_ were calculated with the GraphPad Prism v.9.1.1 (San Diego, CA, USA).

#### 3.2.4. Thiazole Orange Displacement (DNA Intercalation Assay)

A thiazole displacement assay was performed to measure the DNA intercalation activity. A mixture, containing 1 μM of double-stranded calf thymus DNA (recalculated as a concentration of base pairs) and 2 μM of thiazole orange (TO) staining agent dissolved in H_2_O, was used. The fascaplysins were added at concentrations of 0.39–25 μM. The DMSO concentration in the samples was not more than 0.02%. Propidium iodide (PI) was used as an appositive control and taken in concentrations of 0.039–2.5 μM. TO fluorescence was measured 7 min post-incubation at room temperature using a multimodal plate reader TECAN Spark (Männedorf, Switzerland) at the excitation λ = 480 nm. Fluorescence was recorded at λ = 530 nm with the bandwidth 10 nm. The concentration of the drug causing a decrease in TO fluorescence by 50% (EC_50_) was determined using a non-linear regression approximating algorithm, calculated with the GraphPad Prism v.9.1.1 software (San Diego, CA, USA).

#### 3.2.5. Data and Statistical Analysis

Statistical analyses were performed using GraphPad Prism software v.9.1.1 (San Diego, CA, USA). IC_50_ were presented as mean ± standard deviation (SD). The biological experiments were performed in triplicates.

## 4. Conclusions

In conclusion, we developed a protocol that allowed the quaternization of 1-benzoyl-β-carbolines by UV irradiation to the corresponding fascaplysin derivatives under mild conditions. Our method significantly expands the existing options for the synthesis of a wide range of derivatives of this alkaloid. This gives novel opportunities for detailed studies of the structure–activity relationships among these promising physiologically active molecules. The proposed method was used to synthesize both fascaplysin itself and its thermolabile derivatives 9-benzyloxyfascaplysin and 6-*tert*-butylfascaplysin. The attempts to convert 9-benzyloxyfascaplysin to 9-hydroxyfascaplysin, a promising compound for further synthesis of conjugates for targeted drug delivery, indicated the instability of the latter drug. Consequently, hydroxy derivatives of fascaplysin at other positions are required. The synthesis of 6-*tert*-butylfascaplysin allowed, for the very first time, the evaluation of the cytotoxicity of the compound with the skeleton of fascaplysin but with reduced ability to intercalate into DNA. Of note, a rather minor decrease in the cytotoxicity of this 6-*tert*-butylfascaplysin compared to fascaplysin was found. Consequently, the assumption that the DNA intercalation of fascaplysin is a major mechanism of its action must be critically questioned. Our study highlights the demand for further studies to elucidate the subtle mechanisms of the biological activity of this promising marine natural product.

## Data Availability

The original data are available from the corresponding author on request.

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
