# Peer review of "A New Mild Method for Synthesis of Marine Alkaloid Fascaplysin and Its Therapeutically Promising Derivatives"

_marinedrugs, 2023, doi:10.3390/md21080424_

Round 1

Reviewer 1 Report

This MS describes improved synthesis of marine natural product fascaplysin and biological activities of its derivative. The authors improved the previous approach reported by Radchenko et al. by using photoreaction for the final intramolecular N-quarternization reaction. By using this milder method, they succeeded in synthesizing thermolabile fascaplysin derivatives which could not be obtained by the previous method involving thermal reaction. They also found that introduction of a bulky tert-butyl group at 3-position of fascaplysin remarkably reduced intercalating ability compared with parent fascaplysin. Consequently, this reviewer consider that this MS is worth publishing on Marine Drugs. However, revisions of following points are required.

Minor points,

-Page 4, Line 20: compound 5a is not papaverine but papaveraldine.

-Page 4, Line 21: (2-isopropylebenzoyl) should be corrected to (3-isopropoxybenzoyl).

-Synthesis of 1-benzoyl-beta-carbolines, 14-19 and 36-37 (Page 5, Scheme 2, and Page 8, Scheme 7): Zhu et al. reported that use of a co-oxidizer such as H2O2 (1.5 eq.) was required for better yield when using less than stoichiometric amount of I2 (0.8 eq.), however, the condensations of 7a and 25 with acetophenones were all performed with 0.8 eq. of I2 without using any co-oxidizer. As the result, 1-benzoyl-beta-carbolines 14-19 and 36-17 were all obtained in low yields (5~42% yield). It should be stated why the authors avoid using enough amount of I2 or co-oxidizer as H2O2.

-Page 6, Line 7: “phosphorous oxochloride” should be corrected to “phosphorous oxychloride”.

-Page 6, Line 8: “3-nitroethyleneindole” should be corrected to “3-(2-nitroethenyl)indole”

-Page 6, Scheme 4, structure of compound 22: Is the geometry of the double bond of compound 22 (Z-configuration) is correct? Usually, nitro-aldol reaction of aromatic aldehydes gives E-nitrostylenes.

-Page 7, Line 2 “presumably quinoid compound 27”: Experimental procedure employed in the reaction, spectral data, and physical properties (at least mp and HRMS data) should be shown in “Materials and methods” section.

-Throughout the Materials and methods section, consider the significant digits. For example, in “3.1.1 Synthesis of Compound 21 (Page 10, Line 24)”, the weight of substrate 20 is described as 1 g (4.5 mmol), but the wight of the product is written as 1.012 g. Quantities and significant digits should be unified as 1.00 g (4.50 mmol) and 1.01 g.

 Comment: As described in Page 5, Line 11-12, the quarternization may proceed with a radical mechanism. If so, addition of a radical initiator such as AIBN and BPO would be effective in accelerating the reaction.

Author Response

Point 1. Page 4, Line 20: compound 5a is not papaverine but papaveraldine.

Corrected.

Point 2. Page 4, Line 21: (2-isopropylebenzoyl) should be corrected to (3-isopropoxybenzoyl).

Corrected.

Point 3. Synthesis of 1-benzoyl-beta-carbolines, 14-19 and 36-37 (Page 5, Scheme 2, and Page 8, Scheme 7): Zhu et al. reported that use of a co-oxidizer such as H2O2 (1.5 eq.) was required for better yield when using less than stoichiometric amount of I2 (0.8 eq.), however, the condensations of 7a and 25 with acetophenones were all performed with 0.8 eq. of I2 without using any co-oxidizer. As the result, 1-benzoyl-beta-carbolines 14-19 and 36-17 were all obtained in low yields (5~42% yield). It should be stated why the authors avoid using enough amount of I2 or co-oxidizer as H2O2.

Attempts to reproduce of Zhu et al.’s method encountered extremely low yields of target 1-benzoyl-beta-carbolines with the formation of a large number of by-products. Refusal to use H2O2, as in the method of Battini et al. allowed to increase the yields of the target product and reduce the amount of by-products. However, the use of equimolar or excess iodine, as described by Battini et al. led to the formation of a by-product of iodination of 1-benzoyl-beta-carboline at position 6 which was difficult to separate from the target product. Therefore, we used the combined method of Zhu et al. and Battini et al. with iodine deficiency and without H2O2. This explanation has been added to the text of the article.

Point 4. Page 6, Line 7: “phosphorous oxochloride” should be corrected to “phosphorous oxychloride”.

Corrected.

Point 5. Page 6, Line 8: “3-nitroethyleneindole” should be corrected to “3-(2-nitroethenyl)indole”

Corrected.

Point 6. Page 6, Scheme 4, structure of compound 22: Is the geometry of the double bond of compound 22 (Z-configuration) is correct? Usually, nitro-aldol reaction of aromatic aldehydes gives E-nitrostylenes.

Corrected.

Point 7. Page 7, Line 2 “presumably quinoid compound 27”: Experimental procedure employed in the reaction, spectral data, and physical properties (at least mp and HRMS data) should be shown in “Materials and methods” section.

Deprotection procedures and mass spectrum data for product 27 were added.

Point 8. Throughout the Materials and methods section, consider the significant digits. For example, in “3.1.1 Synthesis of Compound 21 (Page 10, Line 24)”, the weight of substrate 20 is described as 1 g (4.5 mmol), but the wight of the product is written as 1.012 g. Quantities and significant digits should be unified as 1.00 g (4.50 mmol) and 1.01 g.

Corrected.

Point 9.  As described in Page 5, Line 11-12, the quarternization may proceed with a radical mechanism. If so, addition of a radical initiator such as AIBN and BPO would be effective in accelerating the reaction.

We tried to use both of these catalysts, but found the appearance of by-products without increasing the yield of the target product. A description of these experiments has been added to Table 1 and the text of the article.

Reviewer 2 Report

The manuscript reports the synthesis of fascaplysin derivatives utilizing a low temperature UV quaternization technique as well as the biological studies of the synthesized fascaplysin derivative. However, the manuscript cannot be accepted for publication at the current state. The main issue is the new synthesis method proposed by the authors has not been reported/investigated thoroughly.

The reaction conditions (Table 3) that were used to achieve the optimization have not been carried out systematically. For example, compound 36 was used in all the studies which did not result in good yield of 4, but out of sudden, the authors used compound 37 and changed the reaction solvent, time, and temperature to get the best yield. Note that compounds 36 and 37 are different as Iodine is a better leaving group when compared to Bromine. The authors should change one parameter at a time in these types of studies. Did the authors try to carry out reaction with compound 37 using the same reaction parameters as compound 36? Did the authors try using compound 36, but use the same reaction conditions as the final entry of Table 3? These are among the unanswered questions in this manuscript.

The authors proposed an unconventional method of stopping the reaction ½ way (30-90 minutes), evaporate the solvent, washed with acetonitrile or benzene, and the washing evaporated, redissolved with acetonitrile, and irradiated with UV again. This step was repeated for another time. This method is not efficient in terms of solvent usage as well as time consuming. I am sure by optimizing the reaction conditions, the reaction can proceed in one go. Note that by stopping the reaction, evaporate the solvent, and followed by washing, it is considered the completion of a reaction, and the authors should report the percentage yield of it (not after repeating the irradiation reaction for another two times!).

Author Response

Point 1. The reaction conditions (Table 3) that were used to achieve the optimization have not been carried out systematically. For example, compound 36 was used in all the studies which did not result in good yield of 4, but out of sudden, the authors used compound 37 and changed the reaction solvent, time, and temperature to get the best yield. Note that compounds 36 and 37 are different as Iodine is a better leaving group when compared to Bromine. The authors should change one parameter at a time in these types of studies. Did the authors try to carry out reaction with compound 37 using the same reaction parameters as compound 36? Did the authors try using compound 36, but use the same reaction conditions as the final entry of Table 3? These are among the unanswered questions in this manuscript.

We presented the results of quaternization of compound 36, since in this case the yields of target derivative 4 were higher compared to prepare of fascaplysin from the analog of 36 - compound 17, shown in Table 1. We agree with the reviewer that these data are redundant and confusing, since the main task of tables 2 and 3 to show the qualitative improvement in the course of the reaction according to the optimized method under the action of UV irradiation in comparison with the previously used high-temperature quaternization procedure. Redundant data has been removed. We are sure that we could also provide clarifications on other about this work.

Point 2. The authors proposed an unconventional method of stopping the reaction ½ way (30-90 minutes), evaporate the solvent, washed with acetonitrile or benzene, and the washing evaporated, redissolved with acetonitrile, and irradiated with UV again. This step was repeated for another time. This method is not efficient in terms of solvent usage as well as time consuming. I am sure by optimizing the reaction conditions, the reaction can proceed in one go. Note that by stopping the reaction, evaporate the solvent, and followed by washing, it is considered the completion of a reaction, and the authors should report the percentage yield of it (not after repeating the irradiation reaction for another two times!).

We tried to carry out UV quaternization of compound 17 at a one time, but an increase in the exposure time by more than three times did not lead to an increase in the yield of the target product. An increase in the irradiation time of compound 36 by a factor of 1.5 resulted in an increase in the yield up to 30%, however, a further increase in the irradiation time did not increase the yield of the product. We attribute the termination of quaternization of 1-benzoyl-beta-carbolines after a certain time to the fact that the resulting product itself effectively absorbs UV radiation at the wavelength at which quaternization occurs. Therefore, for the effective implementation of the transformation, it is necessary to remove the reaction product from the reaction mixture. Considering the significant difference in solubility of target fascaplysins compared to 1-benzoyl-beta-carbolines, we elaborated the protocol for removing the product from the reaction mixture that is very easy to implement, does not require the use of chromatographic methods, and does not leads to loss of product during the isolation step. Therefore, we consider it very technologically advanced.

Reviewer 3 Report

The authors developed in this manuscript a room-temperature photo-cyclization as key step to synthesis fascaplysin, a marine alkaloid with various bioactivities. Owing to the mild reaction conditions, 9-OBn- and 6-tBu-analogues of fascaplysin could be prepared. The bioassay evaluation of the fascaplysin and its 6-tBu-analogue suggest that there may be other mechanism than the impact of DNA intercalation accounting for the cytotoxic effects of fascaplysin-type compounds. I would like to recommend this manuscript be published in Marine Drugs after the revision of the followings:

1. The C=C double bond of compound 22 was drawn as cis-configuration. However, a same reaction was reported to give trans-product (Angew. Chem. Int. Ed. 2015, 54(13), 4032-4035; FR2115089, 1972-08-11; CN108997172A, 2018-12-14).

2. In the caption of scheme 6, the expression of “H2, PtO2, MeOH, r.t. or acetonitrile, 0 °C, 2 h, then r.t., 16 h” is confusing.

3. The C=C double bond of compound 31 was drawn as Z-configuration. Actually, both the E- and Z-isomer of compound 31 are known compounds (Tetrahedron 1994, 50(37), 10955-10962) and the HNMR data of 31 is identical with the those reported for E-isomer but not the Z-isomer.

4. Scheme 4, the yield under the arrow of step d: “28%” → “28-31%”. It seems that the yields were calculated from compound 22. If so, “28-31% in three steps” is better to be shown. Similar case appeared in the yields for compounds 36 and 37 in scheme 7.

5. The calculated HRMS was not corrected. e.g. the m/z for [M+H]+ of compound 30 should be 248.1281, but not 248.1287 which is for [M+H] containing one more electron. Please check all the HRMS data.

6. Section 3.1.8, last line: product 32 → product 35

7. The reaction temperature in section 3.1.9 is 90°C. However, the temperature for this reaction in all the schemes is 110°C. Actually, for this reaction the authors mentioned that they used the conditions in Ref 60. However, Ref 60 used H2O2 as a co-oxidant which was not used in this manuscript. Ref 60 mixed all the substrates and reagents at the beginning, but this manuscript used a different manipulation. Those modifications are better to be mentioned and discussed in the text.

8. The natural Fascaplysin is a chloride salt. According to the synthetic procedure in section 3.1.10, this manuscript prepared the iodide salts of Fascaplysin and its analogue compounds 3 and 4. The iodide anion should be drawn in schemes 3, 5, and 8.

9. The different anions may affect the results of bioassays. It is better to discuss whether the iodide anion would affect the DNA-binding and the cytotoxicity experiment. Or the chloride form of Fascaplysin may also be used in the bioactivity experiments for comparison.

10. Page 3 of the supporting information: the peak at 1.7 ppm is marked as grease. However, the grease peak in CDCl3 usually appears at around 1.25 ppm and 0.9 ppm (Organometallics 2010, 29, 2176–2179). I suggest the 1.7 ppm peak originate from the trace amount of water in the CDCl3 solvent.

Author Response

Point 1. The C=C double bond of compound 22 was drawn as cis-configuration. However, a same reaction was reported to give trans-product (Angew. Chem. Int. Ed. 201554(13), 4032-4035; FR2115089, 1972-08-11; CN108997172A, 2018-12-14).

Corrected.

Point 2.In the caption of scheme 6, the expression of “H2, PtO2, MeOH, r.t. or acetonitrile, 0 °C, 2 h, then r.t., 16 h” is confusing.

The description of scheme 6 has been changed to make it easier to understand.

Point 3. The C=C double bond of compound 31 was drawn as Z-configuration. Actually, both the E- and Z-isomer of compound 31 are known compounds (Tetrahedron 199450(37), 10955-10962) and the HNMR data of 31 is identical with the those reported for E-isomer but not the Z-isomer.

Corrected.

Point 4. Scheme 4, the yield under the arrow of step d: “28%” → “28-31%”. It seems that the yields were calculated from compound 22. If so, “28-31% in three steps” is better to be shown. Similar case appeared in the yields for compounds 36 and 37 in scheme 7.

The scheme 4 has been corrected. Product yields are added at the end of the scheme, not for a specific reaction.

Point 5. The calculated HRMS was not corrected. e.g. the m/z for [M+H]+ of compound 30 should be 248.1281, but not 248.1287 which is for [M+H] containing one more electron. Please check all the HRMS data.

Corrected.

Point 6. Section 3.1.8, last line: product 32 → product 35

Corrected.

Point 7. The reaction temperature in section 3.1.9 is 90°C. However, the temperature for this reaction in all the schemes is 110°C. Actually, for this reaction the authors mentioned that they used the conditions in Ref 60. However, Ref 60 used H2O2 as a co-oxidant which was not used in this manuscript. Ref 60 mixed all the substrates and reagents at the beginning, but this manuscript used a different manipulation. Those modifications are better to be mentioned and discussed in the text.

The temperature has been corrected. Attempts to reproduce of Zhu et al.’s method with using H2O2 encountered extremely low yields of target 1-benzoyl-beta-carbolines with the formation of a large number of by-products. Refusal to use H2O2, as in the method of Battini et al. allowed to increase the yields of the target product and reduce the amount of by-products. However, the use of equimolar or excess iodine, as described by Battini et al. led to the formation of a by-product of iodination of 1-benzoyl-beta-carboline at position 6 which was difficult to separate from the target product. Therefore, we used the combined method of Zhu et al. and Battini et al. with iodine deficiency and without H2O2. This explanation has been added to the text of the article.

Point 8. The natural Fascaplysin is a chloride salt. According to the synthetic procedure in section 3.1.10, this manuscript prepared the iodide salts of Fascaplysin and its analogue compounds 3 and 4. The iodide anion should be drawn in schemes 3, 5, and 8.

The schemes have been corrected. As a result, fascaplysins were obtained not with the counterion iodine, but with chlorine. Description of the conditions under the schemes and in the experimental section has been added.

Point 9. The different anions may affect the results of bioassays. It is better to discuss whether the iodide anion would affect the DNA-binding and the cytotoxicity experiment. Or the chloride form of Fascaplysin may also be used in the bioactivity experiments for comparison.

Fascaplysins with chlorine counterion were used in the biological activity study.

Point 10. Page 3 of the supporting information: the peak at 1.7 ppm is marked as grease. However, the grease peak in CDCl3 usually appears at around 1.25 ppm and 0.9 ppm (Organometallics 201029, 2176–2179). I suggest the 1.7 ppm peak originate from the trace amount of water in the CDCl3 solvent.

Corrected.